# A consistently adaptive trust-region method

**Fadi Hamad**
Department of Industrial Engineering
University of Pittsburgh
Pittsburgh, PA 15261
fah33@pitt.edu

**Oliver Hinder**
Department of Industrial Engineering
University of Pittsburgh
Pittsburgh, PA 15261
ohinder@pitt.edu

## Abstract

Adaptive trust-region methods attempt to maintain strong convergence guarantees without depending on conservative estimates of problem properties such as Lipschitz constants. However, on close inspection, one can show existing adaptive trust-region methods have theoretical guarantees with severely suboptimal dependence on problem properties such as the Lipschitz constant of the Hessian. For example, TRACE developed by Curtis et al. obtains a $O(\Delta_f L^{3/2} \epsilon^{-3/2}) + \tilde{O}(1)$ iteration bound where $L$ is the Lipschitz constant of the Hessian. Compared with the optimal $O(\Delta_f L^{1/2} \epsilon^{-3/2})$ bound this is suboptimal with respect to $L$. We present the first adaptive trust-region method which circumvents this issue and requires at most $O(\Delta_f L^{1/2} \epsilon^{-3/2}) + \tilde{O}(1)$ iterations to find an $\epsilon$-approximate stationary point, matching the optimal iteration bound up to an additive logarithmic term. Our method is a simple variant of a classic trust-region method and in our experiments performs competitively with both ARC and a classical trust-region method.

## 1 Introduction

Second-order methods are known to quickly and accurately solve sparse nonconvex optimization problems that, for example, arise in optimal control [1], truss design [2], AC optimal power flow [3], and PDE constrained optimization [4]. Recently, there has also been a large push to extend second-order methods to tackle machine learning problems by coupling them with carefully designed subproblem solvers [5, 6, 7, 8, 9, 10, 11, 12, 13, 14, 15, 16, 17, 18, 19].

Much of the early theory for second-order methods focused on showing fast local convergence and (eventual) global convergence [20, 21, 22, 23, 24, 25, 26, 27, 28, 29]. These proofs of global convergence, unsatisfactorily, rested on showing at each iteration second-order methods reduced the function value almost as much as gradient descent [22, Theorem 4.5] [30, Theorem 3.2.], this is despite the fact that in practice second-order methods require far fewer iterations. In 2006, Nesterov and Polyak [31] partially resolved this inconsistency by introducing a new second-order method, cubic regularized Newton's method (CRN). Their method can be used to find stationary points of multivariate and possibly nonconvex functions $f : \mathbf{R}^n \to \mathbf{R}$. Their convergence results assumes the optimality gap

$$\Delta_f := f(x_1) - \inf_{x \in \mathbf{R}^n} f(x)$$

is finite and that the Hessian of $f$ is $L$-Lipschitz:

$$\|\boldsymbol{\nabla}^2 f(x) - \boldsymbol{\nabla}^2 f(x')\| \leq L\|x - x'\| \quad \forall x, x' \in \mathbf{R}^n \tag{1}$$

where $\|\cdot\|$ is the spectral norm for matricies and the Euclidean norm for vectors. If $L$ is known they guarantee their algorithm terminates with an $\epsilon$-approximate stationary point:

$$\|\boldsymbol{\nabla} f(x)\| \leq \epsilon$$

36th Conference on Neural Information Processing Systems (NeurIPS 2022).

after at most

$$O(\Delta_f L^{1/2}\epsilon^{-3/2}) \tag{2}$$

iterations. For sufficiently small $\epsilon$, this improves on the classic guarantee that gradient descent terminates after at most $O(\Delta_f S\epsilon^{-2})$ iterations for $S$-smooth functions, thereby partially resolving this inconsistency between theory and practice. Bound (2) is also known to be the best possible for second-order methods [32].

However, CRN only achieves (2) if the Lipschitz constant of the Hessian is known. In practice, we rarely know the Lipschitz constant of the Hessian, and if we do it is likely to be a conservative estimate. With this in mind, many authors have developed practical algorithms that achieve the convergence guarantees of CRN without needing to know the Lipschitz constant of the Hessian. We list these adaptive second-order methods in Table 1 along with their worst-case iteration bounds.

Despite the fact that all these algorithms match the $\epsilon$-dependence of (2), the majority of them are suboptimal due to the dependency on the Lipschitz constant $L$. For example, only our method and [33, 34] are optimal in terms of $L$ scaling. Whereas [35] is suboptimal as the bound scales proportional to $L^{3/2}$ instead of $L^{1/2}$. Moreover, all the trust-region methods have suboptimal $L$ scaling. In particular, inspection of these bounds shows scaling with respect to $L$ of $L^{3/2}$ for [36] and $L^2$ for [37] instead of the optimal scaling of $L^{1/2}$.

An ideal algorithm wouldn't incur this cost for adaptivity. This motivates the following definition.

**Definition 1.** *A method is **consistently adaptive** on a problem class if, without knowing problem parameters, it achieves the same worst-case complexity bound as one obtains if problem parameters were known, up to a **problem-independent** constant-factor and additive polylogarithmic term.*

Clearly, based on our above discussion there does not exist consistently adaptive trust-region methods. Indeed, despite the extensive literature on trust-region methods [9, 10, 22, 25, 27, 36, 38, 39, 40, 41] and their worst-case iterations bounds [36, 37, 42], none of these methods are consistently adaptive. As we mentioned earlier and according to Table 1, [33, 34] are cubic regularization based methods which scale optimally with respect to the problem parameters. However, they are not quite consistently adaptive because $\sigma_0$ appears outside the additive polylogarithmic term.

Table 1: Adaptive second-order methods along with their worst-case bounds on the number of gradient, function and Hessian evaluations. $\sigma_{\min} \in (0, \infty)$ is the smallest regularization parameter used by ARC [35]. $\sigma_0 \in (0, \infty)$ is the initial regularization parameter for cubic regularized methods.

| Algorithm | type | worst-case iterations bound |
|---|---|---|
| **ARC [35]** [1] | cubic regularized | $O(\Delta_f L^{3/2}\sigma_{\min}^{-1}\epsilon^{-3/2} + \Delta_f \sigma_{\min}^{1/2}\epsilon^{-3/2})$ |
| **Nesterov et al. [33, Eq. 5.13 and 5.14]** [2] | cubic regularized | $O(\Delta_f \max\{L, \sigma_0\}^{1/2}\epsilon^{-3/2}) + \tilde{O}(1)$ |
| **ARp [34, Section 4.1]** [3] | cubic regularized | $O(\Delta_f \max\{L, \sigma_0\}^{1/2}\epsilon^{-3/2}) + \tilde{O}(1)$ |
| **TRACE [36, Section 3.2]** | trust-region | $O(\Delta_f L^{3/2}\epsilon^{-3/2}) + \tilde{O}(1)$ |
| **Toint et al. [37, Section 2.2]** | trust-region | $\tilde{O}\left(\Delta_f \max\left\{L^2, 1 + 2L\right\}\epsilon^{-3/2}\right)$ |
| **Our method** | trust-region | $O(\Delta_f L^{1/2}\epsilon^{-3/2}) + \tilde{O}(1)$ |

**Our contributions:**

1. We present the first consistently adaptive trust-region method for finding stationary points of nonconvex functions with $L$-Lipschitz Hessians and bounded optimality gap. In particular, we prove our method finds an $\epsilon$-approximate stationary point after at most $O(\Delta_f L^{1/2} \epsilon^{-3/2}) + \tilde{O}(1)$ iterations.

2. We show our trust-region method has quadratic convergence when it enters a region around a point satisfying the second-order sufficient conditions for local optimality.

3. Our method appears promising in experiments. We test our method on the CUTEst test set [43] against other methods including ARC and a classic trust-region method. These tests show how competitive we are against the other methods in term of total number of required iterations until convergence.

**Paper outline**   The paper is structured as follows. Section 2 presents our trust-region method and contrasts it with existing trust-region methods. Section 3 presents our main result: a convergence bound for finding $\epsilon$-approximate stationary points that is consistently adaptive to problems with Lipschitz continuous Hessian. Section 4 shows quadratic convergence of the method. Section 5 discusses the experimental results.

**Notation**   Let $\mathbf{N}$ be the set of natural numbers (starting from one), $\mathbf{I}$ be the identity matrix, and $\mathbf{R}$ the set of real numbers. Throughout this paper we assume that $n \in \mathbf{N}$ and $f : \mathbf{R}^n \to \mathbf{R}$ is bounded below and twice-differentiable. We define $f_\star := \inf_{x \in \mathbf{R}^n} f(x)$ and $\Delta_f := f(x_1) - f_\star$.

## 2   Our trust-region method

### 2.1   Trust-region subproblems

As is standard for trust-region methods [22] at each iteration $k$ of our algorithm we build a second-order Taylor series approximation at the current iterate $x_k$:

$$M_k(d) := \frac{1}{2} d^T \boldsymbol{\nabla}^2 f(x_k) d + \boldsymbol{\nabla} f(x_k)^T d \tag{3}$$

and minimize that approximation over a ball with radius $r_k > 0$:

$$\min_{d \in \mathbf{R}^n} M_k(d) \text{ s.t. } \|d\| \leq r_k \tag{4}$$

to generate a search direction $d_k$. One important practical question is given a candidate search direction $d_k$, how can we verify that it solves (4). For this one can use the following well-known Fact.

**Fact 1** (Theorem 4.1 [44])**.** *The direction $d_k$ exactly solves* (4) *if and only there exists $\delta_k \in [0, \infty)$ such that:*

$$\boldsymbol{\nabla} M_k(d_k) + \delta_k d_k = 0 \tag{5a}$$

$$\delta_k r_k \leq \delta_k \|d_k\| \tag{5b}$$

$$\|d_k\| \leq r_k \tag{5c}$$

$$\boldsymbol{\nabla}^2 f(x_k) + \delta_k \mathbf{I} \succeq 0 \tag{5d}$$

---

[1]Obtaining this bound does require carefully inspection of Cartis, Gould and Toint [35] (who highlighted only on the $\epsilon$-dependence of their bound). For simplicity of discussion we assume the ARC subproblems are solved exactly (i.e., $C = 0$, $\kappa_\theta = 0$), and that the initial regularization parameter satisfies $\sigma_0 = O(L + \sigma_{\min})$ (the bound only gets worse otherwise). We also consider only the bound on the number of Hessian evaluations, inclusion of the unsuccessful iterations (where cubic regularized subproblems are still solved) makes this bound even worse. Finally, we ignore problem-independent parameters $\gamma_1, \gamma_2, \gamma_3$, and $\eta_1$.

[2]Since by our assumption the function $f$ has $L$-Lipschitz Hessian, we only consider the case when the Hölder exponent $\nu = 1$. Note also that the algorithm description [33, Eq. 5.12], requires that the initial regularization parameter $\sigma_0$ ($H_0$ using their notation) satisfies $H_0 \in (0, H_f(v)]$ where $H_f(v)$ is defined in [33, Eq. 2.1]. Technically this condition is not verifiable as $H_f(v)$ is unknown in practice. However, one can readily modify [33] by redefining $H_f(v)$ to be the maximum of $H_0$ and the RHS of [33, Eq. 2.1] to remove the requirement that $H_0 \in (0, H_f(v)]$. This gives the bound stated in Table 1.

[3]We only consider the case for the cubic regularized model when $p = 2$ and $r = p + 1 = 3$. Also, since by our assumption the function $f$ has $L$-Lipschitz Hessian, we only consider the case when the Hölder exponent $\beta_2 = 1$.

*which solves* (4).

In practice, it is not possible to exactly solve the trust-region subproblem defined in (4), instead we only require that the trust-region subproblem is approximately solved. For our method, it will suffice to find a direction $d_k$ satisfying:

$$\|\boldsymbol{\nabla} M_k(d_k) + \delta_k d_k\| \leq \gamma_1 \|\boldsymbol{\nabla} f(x_k + d_k)\| \tag{6a}$$

$$\gamma_2 \delta_k r_k \leq \delta_k \|d_k\| \tag{6b}$$

$$\|d_k\| \leq r_k \tag{6c}$$

$$M_k(d_k) \leq -\gamma_3 \frac{\delta_k}{2} \|d_k\|^2 \tag{6d}$$

where $\delta_k$ denotes the solution for the above system and $\gamma_1 \in [0, 1), \gamma_2 \in (1/\omega, 1], \gamma_3 \in (0, 1]$. Setting $\gamma_1 = 0, \gamma_2 = 1, \gamma_3 = 1$ represents the exact version of these conditions. As Lemma 1 shows, exactly solving the trust-region subproblem gives a solution to the system (6). However, the converse it not true, an exact solution to (6) does not necessarily solve the trust-region subproblem. Nonetheless, these conditions are all we need to prove our results, and are easier to verify than a relaxation of (4) that includes a requirement like (5d) which needs a computationally expensive eigenvalue calculation.

**Lemma 1.** *Any solution to* (5) *is a solution to* (6) *with* $\gamma_1 = 0, \gamma_2 = 1, \gamma_3 = 1$.

*Proof.* The only tricky part is proving (6d). However, this can be shown using standard arguments: $M_k(d_k) = \frac{1}{2} d_k^T \boldsymbol{\nabla}^2 f(x_k) d_k + \boldsymbol{\nabla} f(x_k)^T d_k = -\frac{1}{2} d_k^T (\boldsymbol{\nabla}^2 f(x_k) + 2\delta_k \mathbf{I}) d_k \leq -\frac{\delta_k}{2} \|d_k\|^2$ where the second equality uses (5a) and the inequality (5d). $\qquad \square$

## 2.2 Our trust-region method

An important component of a trust-region method is the decision for computing the radius $r_k$ at each iteration. This choice is based on whether the observed function value reduction $f(x_k) - f(x_k + d_k)$ is comparable to the predicted reduction from the second-order Taylor series expansion $M_k$. In particular, given a search direction $d_k$ existing trust-region methods compute the ratio

$$\rho_k := \frac{f(x_k) - f(x_k + d_k)}{-M_k(d_k)} \tag{7}$$

and then increase $r_k$ if $\rho_k \geq \beta$ or decrease $r_k$ if $\rho_k < \beta$ [22]. Unfortunately, while intuitive, this criteria is provably bad, in the sense that one can construct examples of functions with Lipschitz continuous Hessians where any trust-region method that uses this criteria will have a convergence rate proportional to $\epsilon^{-2}$ [45, Section 3].

Instead of (7), we introduce a variant of this ratio by adding the term $\frac{\theta}{2} \|\boldsymbol{\nabla} f(x_k + d_k)\| \|d_k\|$ to the predicted reduction where $\theta \in (0, \infty)$ is a problem-independent hyperparameter (we use $\theta = 0.1$ in our implementation). This requires the algorithm to reduce the function value more if the gradient norm at the candidate solution $x_k + d_k$, and search direction norm are big. In particular, we define our new ratio as:

$$\hat{\rho}_k := \frac{f(x_k) - f(x_k + d_k)}{-M_k(d_k) + \frac{\theta}{2} \|\boldsymbol{\nabla} f(x_k + d_k)\| \|d_k\|} \tag{8}$$

Our trust-region method is presented in Algorithm 1. The algorithm includes some other minor modification of classic trust-region methods [22]: we accept all search directions that reduce the function value, and update the $r_{k+1}$ using $\|d_k\|$ instead of $r_k$ (see [46, Equation 13.6.13] for a similar update rule). We recommend contrasting our algorithm with [36, Algorithm 1] which is trust-region method with an iteration bound proportional to $\epsilon^{-3/2}$ but is more complex and not consistently adaptive.

For the remainder of this paper $x_k$ and $d_k$ refer to the iterates of Algorithm 1.

## 3 Proof of full adaptivity on Lipschitz continuous functions

This section proves that our method is consistently adaptive for finding approximate stationary points on functions with $L$-Lipschitz Hessians. The core idea behind our proof is to get a handle on the size

**Algorithm 1:** Consistently Adaptive Trust Region Method (CAT)

---

**Input requirements:** $r_1 \in (0, \infty)$, $x_1 \in \mathbf{R}^n$ ;
**Problem-independent parameter requirements:** $\theta \in (0,1), \beta \in (0,1), \omega \in (1, \infty)$,
$\gamma_1 \in [0,1), \gamma_2 \in (1/\omega, 1], \gamma_3 \in (0,1], \frac{\beta\theta}{\gamma_3(1-\beta)} + \gamma_1 < 1$ ;
**for** $k = 1, \ldots, \infty$ **do**

> Approximately solve the trust-region subproblem, i.e., obtain $d_k$ that satisfies (6) ;
>
> $x_{k+1} \leftarrow \begin{cases} x_k + d_k & f(x_k + d_k) \leq f(x_k) \\ x_k & \text{otherwise} \end{cases}$
>
> $r_{k+1} \leftarrow \begin{cases} \omega\|d_k\| & \hat{\rho}_k \geq \beta \\ \|d_k\|/\omega & \text{otherwise} \end{cases}$

---

of $\|d_k\|$. In particular, if we can bound $\|d_k\|$ from below and $\hat{\rho}_k \geq \beta$ then the $\frac{\theta}{2}\|\boldsymbol{\nabla} f(x_k + d_k)\|\|d_k\|$ term guarantees that at iteration $k$ the function value is reduced by a large amount relative to the gradient norm $\|\boldsymbol{\nabla} f(x_k + d_k)\|$.

Lemma 2 guarantees the norm of the gradient for the candidate solution $x_k + d_k$ lower bounds the size of $\|d_k\|$ under certain conditions. Note this bound on the gradient, i.e., (11) holds without us needing to know the Lipschitz constant of the Hessian $L$. The proof of Lemma 2 appears in Section A.1 and heavily leverages Fact 2.

**Fact 2** (Nesterov & Polyak 2006, Lemma 1 [31]). *If $\boldsymbol{\nabla}^2 f$ is L-Lipschitz,*

$$\|\boldsymbol{\nabla} f(x_k + d_k)\| \leq \|\boldsymbol{\nabla} M_k(d_k)\| + \frac{L}{2}\|d_k\|^2 \tag{9}$$

$$f(x_k + d_k) \leq f(x_k) + M_k(d_k) + \frac{L}{6}\|d_k\|^3. \tag{10}$$

**Lemma 2.** *Suppose $\boldsymbol{\nabla}^2 f$ is L-Lipschitz. If $\|d_k\| < \gamma_2 r_k$ or $\hat{\rho}_k \leq \beta$ then*

$$\|\boldsymbol{\nabla} f(x_k + d_k)\| \leq c_1 L\|d_k\|^2 \tag{11}$$

*where $c_1 > 0$ is a problem-independent constant:*

$$c_1 := \max\left\{\frac{5 - 3\beta}{6(\gamma_3(1 - \gamma_1)(1 - \beta) - \beta\theta)}, \frac{1}{2(1 - \gamma_1)}\right\}.$$

For the remainder of this section we will find the following quantities useful,

$$\underline{d}_\epsilon := \gamma_2\omega^{-1}c_1^{-1/2}L^{-1/2}\epsilon^{1/2}$$

$$\bar{d}_\epsilon := \frac{2\omega}{\beta\theta} \cdot \frac{\Delta_f}{\epsilon}.$$

As we will show shortly in Lemma 4, after a short warm up period $\underline{d}_\epsilon$ and $\bar{d}_\epsilon$ represent lower and upper bound on $\|d_k\|$ (i.e., $\underline{d}_\epsilon \leq \|d_k\| \leq \bar{d}_\epsilon$) as long as $\|\boldsymbol{\nabla} f(x_k + d_k)\| \geq \epsilon$. But before presenting and proving Lemma 4 we develop Lemma 3 which is a stepping stone to proving Lemma 4. Lemma 3 shows that if $\|d_k\|$ is almost above $\bar{d}_\epsilon$ then the trust-region radius will shrink, and if $\|d_k\|$ is almost below $\underline{d}_\epsilon$ then the trust-region radius will grow (recall from Algorithm 1 that $\omega \in (1, \infty)$).

**Lemma 3.** *Suppose $\boldsymbol{\nabla}^2 f$ is L-Lipschitz. Let $\epsilon \in (0, \infty)$ and $\|\boldsymbol{\nabla} f(x_k + d_k)\| \geq \epsilon$ then*

1. *If $\|d_k\| > \bar{d}_\epsilon/\omega$ then $\|d_k\|/\omega = r_{k+1}$.*

2. *If $\|d_k\| < \omega\gamma_2^{-1}\underline{d}_\epsilon$ then $\gamma_2 r_k \leq \|d_k\| \leq r_k$ & $\omega\|d_k\| = r_{k+1}$.*

*Proof.* Proof for 1. We have

$$\|d_k\| > \bar{d}_\epsilon/\omega = 2\frac{\Delta_f}{\beta\theta\epsilon} \geq 2\frac{f(x_k) - f(x_{k+1})}{\beta\theta\epsilon} \tag{12}$$

where the first equality uses the definition of $\bar{d}_\epsilon$ and the second inequality uses $f(x_{k+1}) \geq \inf_{x \in \mathbf{R}^n} f(x)$ and $f(x_1) \geq f(x_k)$. Furthermore,

$$\hat{\rho}_k = \frac{f(x_k) - f(x_k + d_k)}{-M_k(d_k) + \frac{\theta}{2}\|\boldsymbol{\nabla} f(x_k + d_k)\|\|d_k\|} \leq \frac{f(x_k) - f(x_{k+1})}{\frac{\theta}{2}\|\boldsymbol{\nabla} f(x_k + d_k)\|\|d_k\|} \leq 2\frac{f(x_k) - f(x_{k+1})}{\theta \epsilon \|d_k\|} < \beta$$

where the first inequality follows from $-M_k(d_k) \geq 0$ and $f(x_{k+1}) \leq f(x_k + d_k)$, the second inequality follows from the fact that $\|\boldsymbol{\nabla} f(x_k + d_k)\| \geq \epsilon$, and the third inequality uses (12). By inspection of Algorithm 1, if $\hat{\rho}_k < \beta$, then $\|d_k\|/\omega = r_{k+1}$.

Proof for 2. We will prove the result by contrapositive. In particular, suppose that $\neg(\gamma_2 r_k \leq \|d_k\| \leq r_k)$ or $\|d_k\|\omega \neq r_{k+1}$. Let us consider these two cases. If $\neg(\gamma_2 r_k \leq \|d_k\| \leq r_k)$ then as $\|d_k\| \leq r_k$ we have $\gamma_2 r_k > \|d_k\|$. If $\|d_k\|\omega \neq r_{k+1}$ then by inspection of Algorithm 1 we have $\hat{\rho}_k \leq \beta$. Therefore, in both these cases the premise of Lemma 2 holds. Now, by $\|\boldsymbol{\nabla} f(x_k + d_k)\| \geq \epsilon$ and Lemma 2 we get

$$\epsilon \leq \|\boldsymbol{\nabla} f(x_k + d_k)\| \leq c_1 L \|d_k\|^2$$

which implies $\|d_k\| \geq c_1^{-1/2} L^{-1/2} \epsilon^{1/2} = \gamma_2^{-1}\omega(\gamma_2 \omega^{-1} c_1^{-1/2} L^{-1/2} \epsilon^{1/2}) = \gamma_2^{-1}\omega \underline{d}_\epsilon$. $\qquad\square$

Now we show that the norm of the direction $d_k$, after some finite iteration $\underline{k}_\epsilon$, will be bounded below and above by $\underline{d}_\epsilon$ and $\bar{d}_\epsilon$ respectively. For that we first define:

$$K_\epsilon := \min\{\{k \in \mathbf{N} : \|\boldsymbol{\nabla} f(x_k + d_k)\| \leq \epsilon\} \cup \{\infty\}\}$$

as the first iteration for which $\|\boldsymbol{\nabla} f(x_k + d_k)\| \leq \epsilon$, and we also define:

$$\underline{k}_\epsilon := \min\{\{k \in \mathbf{N} : \underline{d}_\epsilon \leq \|d_k\| \leq \bar{d}_\epsilon\} \cup \{K_\epsilon - 1\}\}$$

as the first iteration for which $\underline{d}_\epsilon \leq \|d_k\| \leq \bar{d}_\epsilon$. An illustration of Lemma 4 is given in Figure 1. In particular, after a certain warm up period the direction norms can no longer rise above $\bar{d}_\epsilon$ or below $\underline{d}_\epsilon$. Broadly speaking, the idea behind the proof is that if $\|d_k\|$ is above $\bar{d}_\epsilon/\omega$ then at the next iteration $\|d_k\|$ decreases and conversely if $\|d_k\|$ is bellow $\underline{d}_\epsilon \omega \gamma_2^{-1}$ then at the next iteration it must increase.

**Lemma 4.** *Suppose $\boldsymbol{\nabla}^2 f$ is L-Lipschitz and let $\epsilon \in (0, \infty)$. If $\underline{d}_\epsilon \leq \|d_k\| \leq \bar{d}_\epsilon$ then $\underline{d}_\epsilon \leq \|d_j\| \leq \bar{d}_\epsilon$ for all $j \in [k, K_\epsilon) \cap \mathbf{N}$. Furthermore, $\underline{k}_\epsilon \leq 1 + \log_{\gamma_2 \omega}(\max\{1, \underline{d}_\epsilon/r_1, r_1/\bar{d}_\epsilon\})$.*

*Proof.* We begin by proving $\underline{d}_\epsilon \leq \|d_k\| \leq \bar{d}_\epsilon$ then $\underline{d}_\epsilon \leq \|d_j\| \leq \bar{d}_\epsilon$ for $j \in [k, K_\epsilon) \cap \mathbf{N}$. We assume that $k < K_\epsilon$ otherwise our desired conclusion clearly holds. We split this proof into two claims.

Our first claim is that $\|d_k\| \leq \bar{d}_\epsilon$ implies $\|d_{k+1}\| \leq \bar{d}_\epsilon$. We split $\|d_k\| \leq \bar{d}_\epsilon$ into two subcases. If $\|d_k\| \leq \bar{d}_\epsilon/\omega$, then inspection of Algorithm 1 shows that $\|d_{k+1}\| \leq r_{k+1} \leq \|d_k\|\omega \leq \bar{d}_\epsilon$. If $\bar{d}_\epsilon/\omega \leq \|d_k\| \leq \bar{d}_\epsilon$, then Lemma 3.1 implies that $\|d_{k+1}\| \leq r_{k+1} \leq \|d_k\|/\omega \leq \bar{d}_\epsilon$.

Our second claim is that $\underline{d}_\epsilon \leq \|d_k\|$ implies $\underline{d}_\epsilon \leq \|d_{k+1}\|$. We split $\underline{d}_\epsilon \leq \|d_k\|$ into three subcases. If $\|d_{k+1}\| < \gamma_2 r_{k+1}$, then the contrapositive of Lemma 3.2 implies that $\|d_{k+1}\| \geq \underline{d}_\epsilon$. If $\gamma_2 r_{k+1} \leq \|d_{k+1}\| \leq r_{k+1}$ and $\underline{d}_\epsilon \leq \|d_k\| < \gamma_2^{-1}\omega \underline{d}_\epsilon$, then $\underline{d}_\epsilon < \gamma_2 \omega \underline{d}_\epsilon \leq \gamma_2 \omega \|d_k\| =_\star \gamma_2 r_{k+1} \leq \|d_{k+1}\|$ where $\star$ uses Lemma 3.2. If $\gamma_2 r_{k+1} \leq \|d_{k+1}\| \leq r_{k+1}$ and $\underline{d}_\epsilon \omega \gamma_2^{-1} \leq \|d_k\|$, then $\underline{d}_\epsilon \leq \frac{\gamma_2 \|d_k\|}{\omega} \leq_\star \gamma_2 r_{k+1} \leq \|d_{k+1}\|$ where $\star$ is from the update rule for $r_{k+1}$ in Algorithm 1.

By induction on the previous two claims we deduce if $\underline{d}_\epsilon \leq \|d_k\| \leq \bar{d}_\epsilon$ then $\underline{d}_\epsilon \leq \|d_j\| \leq \bar{d}_\epsilon$ for $j \in [k, K_\epsilon) \cap \mathbf{N}$.

Next, we prove that if

$$k \geq 1 + \log_{\omega \gamma_2}(\underline{d}_\epsilon/r_1)$$

then $\|d_k\| \geq \underline{d}_\epsilon$. If $\underline{d}_\epsilon \leq \|d_j\|$ for some $j \leq k$, then the result holds because as we already established $\underline{d}_\epsilon \leq \|d_k\| \Rightarrow \underline{d}_\epsilon \leq \|d_{k+1}\|$. On the other hand, if $\|d_j\| < \underline{d}_\epsilon$ for all $j \leq k$, then Lemma 3.2 implies that $r_{j+1} = \omega\|d_j\| \geq \omega \gamma_2 r_j$ which by induction gives $\|d_k\| \geq (\omega \gamma_2)^{k-1} r_1 \geq (\omega \gamma_2)^{\log_{\omega \gamma_2}(\underline{d}_\epsilon/r_1)} r_1 = \underline{d}_\epsilon$.

Finally, we prove that if

$$k \geq 1 + \log_\omega(r_1/\bar{d}_\epsilon)$$

then $\|d_k\| \leq \bar{d}_\epsilon$. If $\|d_j\| \leq \bar{d}_\epsilon$ for some $j \leq k$, then the result holds because as we already established $\|d_k\| \leq \bar{d}_\epsilon \Rightarrow \|d_{k+1}\| \leq \bar{d}_\epsilon$. On the other hand, if $\|d_j\| > \bar{d}_\epsilon$ for all $j \leq k$, then Lemma 3.1 implies

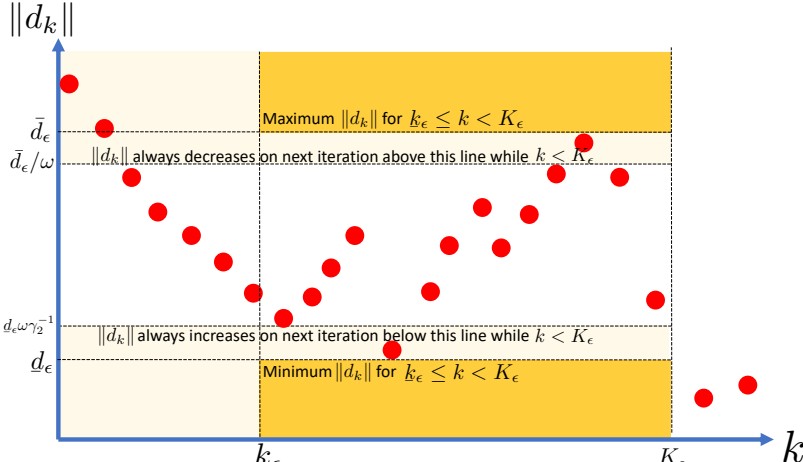

Figure 1: An example of a plausible sequence of iterates and the norms of their directions. Each red dot represents an iterate and its search direction norm. This illustrates Lemma 4.

that $r_{j+1} = \|d_j\|/\omega \leq r_j/\omega$ which by induction gives $\|d_k\| \leq \omega^{1-k} r_1 \leq \omega^{-\log_\omega(r_1/\bar{d}_\epsilon)} r_1 = \bar{d}_\epsilon$. $\quad\square$

Let

$$\mathcal{P}_\epsilon := \{k \in \mathbf{N} : \hat{\rho}_k \geq \beta, \underline{k}_\epsilon \leq k < K_\epsilon\}$$

which represents the set of iterations, before we find an $\epsilon$-approximate stationary point, where the function value is reduced a large amount compared with our target reduction, i.e., $\hat{\rho}_k \geq \beta$. Lemma 5 shows that there is a finite number of these iterations until the gradient drops below the target threshold $\epsilon$. The proof of Lemma 5 appears in Appendix A.2. Roughly, the idea of the proof is to use that, due our definition of $\hat{\rho}_k$, when $\hat{\rho}_k \geq \beta$ we always reduce the function value by at least $\frac{\beta\theta}{2}\|\nabla f(x_k + d_k)\|\|d_k\|$ and $\|d_k\|$ can be lower bounded by $\underline{d}_\epsilon$ using Lemma 4. As we cannot reduce the function value by a constant value indefinitely, we must eventually have $\|\nabla f(x_k + d_k)\| \leq \epsilon$.

**Lemma 5.** *Suppose $\nabla^2 f$ is $L$-Lipschitz and $\epsilon \in (0, \infty)$ then $|\mathcal{P}_\epsilon| \leq \frac{\bar{d}_\epsilon}{\underline{d}_\epsilon\omega} + 1 = \frac{2\omega c_1^{1/2}}{\beta\theta} \cdot \frac{\Delta_f L^{1/2}}{\epsilon^{-3/2}} + 1$.*

With Lemma 5 in hand we are now ready to prove our main result, Theorem 1. We have already provided a bound on the length of the warm up period, $\underline{k}_\epsilon$ (Lemma 4) and on the number of points with $\hat{\rho}_k \geq \beta$. Therefore, the only obstacle is to bound the number of points with $\hat{\rho}_k < \beta$. However, on these iterations we always decrease the radius by at least $\omega$ (see update rules in Algorithm 1), and therefore as $\|d_k\|$ is bounded below by $\underline{d}_\epsilon$, there must be iterations where we increase the radius $r_k$, which by definition of Algorithm 1 only occurs if $\beta \geq \hat{\rho}_k$. Consequently, the number of iterations where $\beta < \hat{\rho}_k$ can be bounded by the number of iterations where $\beta \geq \hat{\rho}_k$ plus a $\tilde{O}(1)$ term. This is the crux of the proof of Theorem 1 which appears in Section A.3.

**Theorem 1.** *Suppose that $\nabla^2 f$ is $L$-Lipschitz and $f$ is bounded below with $\Delta_f = f(x_1) - f_\star$, then for all $\epsilon \in (0, \infty)$ there exists some iteration $k$ with $\|\nabla f(x_k + d_k)\| \leq \epsilon$ and*

$$k \leq O\left(\frac{\Delta_f L^{\frac{1}{2}}}{\epsilon^{\frac{3}{2}}} + \log\left(\frac{\epsilon^{\frac{1}{2}}}{L^{\frac{1}{2}} r_1} + \frac{r_1 \epsilon}{\Delta_f} + 1\right) + 1\right)$$

*where $O(\cdot)$ hides problem-independent constant factors and $r_1$ is the initial trust-region radius.*

One drawback of Theorem 1 is that it only bounds the number of iterations to find a first-order stationary point. Many second-order methods in the literature show convergence to points satisfying the second-order optimality conditions [47, 35, 36, 18]. Of course, these methods are not consistently adaptive. Therefore, in the future, it would be interesting to develop a method that provides a consistently adaptive convergence guarantee for finding second-order stationary points.

# 4 Quadratic convergence when sufficient conditions for local optimality hold

**Theorem 2.** *Suppose $f$ is twice differentiable and for some $x_\star \in R^n$ the second-order sufficient conditions for local optimality hold ($\nabla f(x_\star) = 0$ and $\nabla^2 f(x_\star) \succ 0$). Under these conditions there exists a neighborhood $N$ around $x_\star$ and a constant $c > 0$ such that if $x_i \in N$ then there exist $x_{\hat{k}} \in N$ such that for all $k \geq \hat{k}$ we have $\|x_{k+1} - x_\star\| \leq c\|x_k - x_\star\|^2 \leq \frac{1}{2}\|x_k - x_\star\|$.*

The proof of Theorem 2 appears in Section B. It is a little tricker than typical quadratic convergence proofs for trust-region methods because in our method we have $\lim_{k\to 0} r_k \to 0$ whereas classical trust-region methods have $r_k$ bounded away from zero [22, Proof of Theorem 4.14]. Fortunately, one can show that asymptotically $r_k \geq \omega\|x_k - x_\star\|$ so the decaying radius does not interfere with quadratic convergence. In particular, the crux of proving Theorem 2 is proving the premise of Lemma 6 (as the conclusion of Lemma 6 is $r_k \geq \omega\|x_k - x_\star\|$). For this Lemma we define $\mathbf{diam}(X) := \sup_{x,x' \in X} \|x - x'\|$.

**Lemma 6.** *Let $N$ be a bounded set such that for all $x_k \in N$ we have $x_{k+1} \in N$, $\hat{\rho}_k \geq \beta$, and $\min\{\gamma_2 r_k, \|x_{k+1} - x_\star\|\} \leq \|d_k\| \leq \omega\gamma_2\|x_k - x_\star\|$. Suppose there exists $x_k \in N$ and let $i$ be the smallest index with $x_i \in N$, then $r_k \geq \omega\|x_k - x_\star\|$ for all $k \geq 2 + i + \log_{\gamma_2\omega}(\frac{\mathbf{diam}(N)}{\|d_i\|})$.*

*Proof.* Let $k \geq i$. By induction $x_k \in N$. By $\hat{\rho}_k \geq \beta$ and inspection of Algorithm 1 we have $r_{k+1} = \omega\|d_k\|$. Suppose that $\|d_k\| \geq \gamma_2 r_k$ for all $i \leq k \leq i + \ell$ then $\mathbf{diam}(N) \geq \|d_{k+1}\| \geq \gamma_2 r_{k+1} = \gamma_2\omega\|d_k\| = \gamma_2^\ell\omega^\ell\|d_i\|$. Rearranging gives $\ell \leq \log_{\gamma_2\omega}(\frac{\mathbf{diam}(N)}{\|d_i\|})$. Next observe that if $\|d_k\| < \gamma_2 r_k$ then $\|d_{k+1}\| \leq \omega\gamma_2\|x_{k+1} - x_\star\| \leq \omega\gamma_2\|d_k\| = (\omega\gamma_2/\omega)r_{k+1} = r_{k+1}$. By induction $\|d_k\| < \gamma_2 r_k$ for all $k > \ell$. Finally, observe that if $\|d_k\| < \gamma_2 r_k$ then $r_{k+1} = \omega\|d_k\| \geq \omega\|x_{k+1} - x_\star\|$. $\square$

# 5 Experimental results

We test our algorithm on learning linear dynamical systems [48], matrix completion [49], and the CUTEst test set [50]. Appendix D contains the complete set of results from our experiments. Our method is implemented in an open-source Julia module available at `https://github.com/fadihamad94/CAT-NeurIPS`. The implementation uses a factorization and eigendecomposition approach to solve the trust-region subproblems (i.e., satisfy (6)). We perform our experiments using Julia 1.6 on a Linux virtual machine that has 8 CPUs and 16 GB RAM. The CAT code repository provides instructions for reproducing the experiments and detailed tables of results.

For these experiments, the selection of the parameters (unless otherwise specified) is as follow: $r_1 = 1.0$, $\beta = 0.1$, $\theta = 0.1$, $\omega = 8.0$, $\gamma_1 = 0.0$, $\gamma_2 = 0.8$, and $\gamma_3 = 1.0$. When implementing Algorithm 1 with some target tolerance $\epsilon$, we immediately terminate when we observe a point $x_k$ with $\|\nabla f(x_k + d_k)\| \leq \epsilon$. This also includes the case when we check the inner termination criteria for the trust-region subproblem. The full details of the implementation are described in Appendix C.

## 5.1 Learning linear dynamical systems

We test our method on learning linear dynamical systems [48] to see how efficient our method compared to a trust-region solver. We synthetically generate an example with noise both in the observations and also the evolution of the system, and then recover the parameters using maximum likelihood estimation. Details are provided in Appendix D.1.

On this problem we compare our algorithm with a Newton trust-region method that is available through the Optim.jl package [51]. The comparisons are summarized in Table 2.

## 5.2 Matrix completion

We also demonstrate the effectiveness of our algorithm against a trust-region solver on the matrix completion problem. The matrix completion formulation can be written as the regularized squared error function of SVD model [49, Equation 10]. For our experiment, we use the public data set of Ausgrid, but we only use the data from a single substation. Details are provided in Appendix D.2.

Again we compare our algorithm with a Newton trust-region method [51]. The comparison are summarized in Table 3.

Table 2: Geometric mean for total number of iterations, and evaluations of the function and gradient, per solver on 60 randomly generated instances with $\epsilon = 10^{-5}$ termination tolerance. Failures counted as the maximum number of iterations (10000) when computing the geometric mean.

|  | #iter | #function | #gradient |
|---|---|---|---|
| **Newton trust-region [51]** | 480.1 | 482.3 | 482.3 |
| **Our method** | 308.1 | 309.6 | 309.6 |
| 95% **CI for ratio** | [1.24, 1.95] | [1.24, 1.94] | [1.24, 1.94] |

Table 3: Geometric mean for total number of iterations per solver on 10 instances by randomly generating the sampled measurements from the matrix D using data from Ausgrid with $\epsilon = 10^{-5}$ termination tolerance. Failures counted as the maximum number of iterations (1000) when computing the geometric mean.

|  | #iter |
|---|---|
| **Newton trust-region [51]** | 1000 |
| **Our method** | 216.4 |

## 5.3 Results on CUTEst test set

The CUTEst test set [50] is a standard test set for nonlinear optimization algorithms. To run the benchmarks we use `https://github.com/JuliaSmoothOptimizers/CUTEst.jl` (the License can be found at `https://github.com/JuliaSmoothOptimizers/CUTEst.jl/blob/main/LICENSE.md`). We will be comparing with the results for ARC reported in [52, Table 1]. As the benchmark CUTEst that we used has changed since [52] was written we select only the problems in CUTEst that remain the same (some of the problem sizes have changed). This gives 67 instances. A table with our full results can be found in the results/CUTEst subdirectory in the git repository for this paper. Catris et.al [52] report the results for three different implementations of their ARC algorithm. We limit our comparison to the ARC g-rule algorithm since it performs better than the other ARC approaches. We also run the Newton trust-region method from the Optim.jl package [51].

Our algorithm is stopped as soon $\|\nabla f(x_k + d_k)\|$ is smaller than $10^{-5}$. For the Newton trust-region method [51] we also used as a stopping criteria a value of $10^{-5}$ for the gradient termination tolerance. We used 10000 as an iteration limit and any run exceeding this is considered a failure. This choice of parameters is to be consistent with [52].

As we can see from Table 4, our algorithm offers significant promise, requiring similar function evaluations (and therefore subproblem solves) to converge than the Newton trust-region of [51] and ARC, although the number of gradient evaluations is slightly higher than ARC. In addition, the comparison between these algorithms in term of total number of iterations and total number of gradient evaluations is summarized in Figure 2.

## 5.4 Convergence of our method with different theta values

To demonstrate the role of the additional $\frac{\theta}{2}\|d_k\|\|\nabla f(x_k + d_k)\|$ term we run experiments with different $\theta$ values. In particular, we contrast our default value of $\theta = 0.1$ with $\theta = 0$ which corresponds to not adding the $\frac{\theta}{2}\|d_k\|\|\nabla f(x_k + d_k)\|$ term to $\hat{\rho}_k$ (recall the discussion in Section 2.2).

Table 4: Number of failures, geometric mean for total number of iterations and function and gradient evaluations per solver on 67 unconstrained instance from the CUTEst benchmark instances. Failures counted as the maximum number of iterations (10000) when computing the geometric mean.

|  | #failures | #iter | #function | #gradient |
|---|---|---|---|---|
| **Our method** | 3 | 41.5 | 44.4 | 44.4 |
| **ARC with the g-rule [52]** | 1 | 38.1 | 38.1 | 26.6 |
| **Newton trust-region [51]** | 4 | 44.5 | 47.2 | 47.2 |

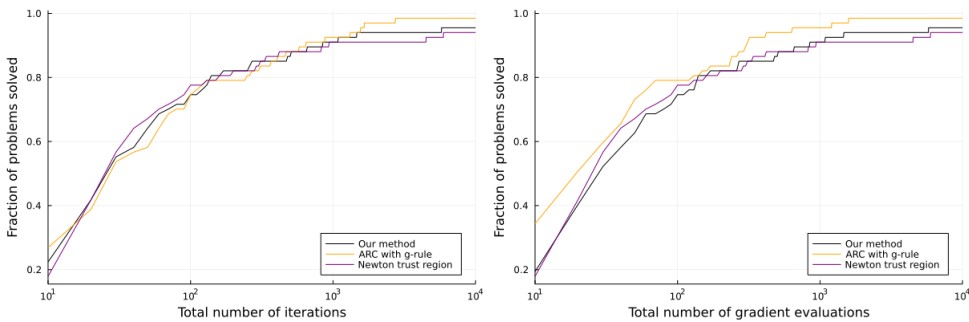

Figure 2: Fraction of problems solved on the CUTEst benchmark.

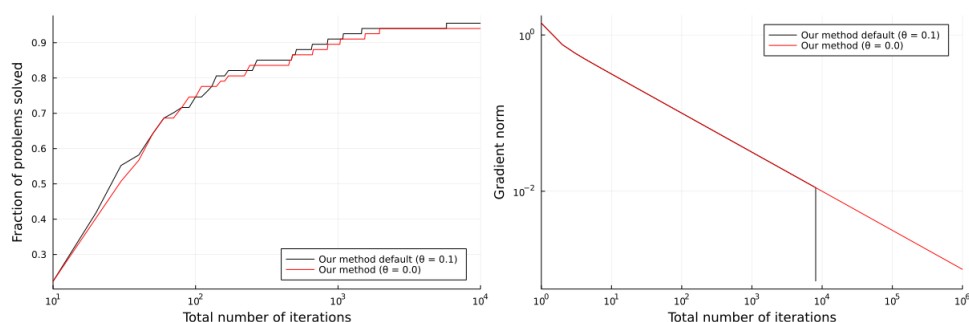

(a) Fraction of problems solved versus total number of iterations on the CUTEst test set

(b) Gradient norm versus total number of iterations on the example of [45] with $r_1 = 1.5$

Figure 3: Performance of our algorithm based on different $\theta$ values.

In Figure 3a we rerun on the CUTEst test set (as per Section 5.3) and compare these two options. One can see the algorithm performs similarly with either $\theta = 0$ or $\theta = 0.1$.

In Figure 3b we test on the hard example from [45] which is designed to exhibit the poor worst-case complexity of trust-region methods (i.e., a convergence rate proportional to $\epsilon^{-2}$) if the initial radius $r_1$ is chosen sufficiently large (to achieve this we set $r_1 = 1.5$). We run this example with $\epsilon = 10^{-3}$. One can see that while for the first $\approx 10^4$ iterations the methods follow identical trajectories, thereafter $\theta = 0.1$ rapidly finds a stationary point whereas $\theta = 0.0$ requires two orders of magnitude more iterations to terminate. This crystallizes the importance of $\theta$ in circumventing the worst-case $\epsilon^{-2}$ convergence rate of trust-region methods.

## Acknowledgments and Disclosure of Funding

The authors were supported by the Pitt Momentum Funds. The authors would like to thank Coralia Cartis and Nicholas Gould for their helpful feedback on a draft of this paper. The authors would also like to thank Xiaoyi Qu for finding errors in the proof of an early draft of the paper and for helpful discussions.

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
