# A Proof of results from Section 3

## A.1 Proof of Lemma 2

*Proof.* First we prove result in the case that $\|d_k\| < \gamma_2 r_k$. By (6b) the statement $\|d_k\| < \gamma_2 r_k$ implies $\delta_k = 0$. Combining $\delta_k = 0$ with (6a) and (9) and using the fact $1 - \gamma_1 > 0$ yields

$$\|\nabla f(x_k + d_k)\| \leq \frac{L}{2(1 - \gamma_1)} \|d_k\|^2 \leq c_1 L \|d_k\|^2 .$$

Next we prove the result in the case that $\hat{\rho}_k \leq \beta$. Then

$$M_k(d_k) + \frac{L}{6} \|d_k\|^3 \geq f(x_k + d_k) - f(x_k) = -\hat{\rho}_k \left( -M_k(d_k) + \frac{\theta}{2} \|\nabla f(x_k + d_k)\| \|d_k\| \right)$$

$$\geq -\beta \left( -M_k(d_k) + \frac{\theta}{2} \|\nabla f(x_k + d_k)\| \|d_k\| \right)$$

where the the first inequality uses (10), the first equality uses the definition of $\hat{\rho}_k$, and the second inequality uses $\hat{\rho}_k \leq \beta$ and $-M_k(d_k) + \frac{\theta}{2} \|\nabla f(x_k + d_k)\| \|d_k\| \geq 0$.

Rearranging the previous inequality using $1 - \beta > 0$ and then applying (6d) yields:

$$\frac{L}{3(1 - \beta)} \|d_k\|^2 + \frac{\beta\theta}{1 - \beta} \|\nabla f(x_k + d_k)\| \geq -\frac{2M_k(d_k)}{\|d_k\|} \geq \gamma_3 \delta_k \|d_k\|. \tag{13}$$

Now, by (9), (6a) and the triangle inequality, and (13) respectively:

$$\|\nabla f(x_k + d_k)\| \leq \|\nabla M_k(d_k)\| + \frac{L}{2} \|d_k\|^2 \leq \delta_k \|d_k\| + \gamma_1 \|\nabla f(x_k + d_k)\| + \frac{L}{2} \|d_k\|^2$$

$$\leq L \left( \frac{1}{3\gamma_3(1 - \beta)} + \frac{1}{2} \right) \|d_k\|^2 + \left( \frac{\beta\theta}{\gamma_3(1 - \beta)} + \gamma_1 \right) \|\nabla f(x_k + d_k)\|.$$

Rearranging the latter inequality for $\|\nabla f(x_k + d_k)\|$ and using $\frac{\beta\theta}{\gamma_3(1-\beta)} + \gamma_1 < 1$ from the requirements of Algorithm 1 yields:

$$\|\nabla f(x_k + d_k)\| \leq \frac{\frac{1}{3\gamma_3(1-\beta)} + \frac{1}{2}}{1 - \frac{\beta\theta}{\gamma_3(1-\beta)} - \gamma_1} L \|d_k\|^2 = \frac{2 + 3\gamma_3(1 - \beta)}{6(\gamma_3(1 - \gamma_1)(1 - \beta) - \beta\theta)} L \|d_k\|^2$$

$$\leq \frac{5 - 3\beta}{6(\gamma_3(1 - \gamma_1)(1 - \beta) - \beta\theta)} L \|d_k\|^2.$$

$\square$

## A.2 Proof of Lemma 5

*Proof.* For conciseness let $m = |\mathcal{P}_\epsilon|$. Suppose that the indices of $\mathcal{P}_\epsilon$ are ordered increasing value by a permutation function $\pi$, i.e., $\mathcal{P}_\epsilon = \{\pi(i) : i \in [m]\}$ with $\pi(1) < \cdots < \pi(m)$. Then

$$\Delta_f \geq f(x_{\pi(1)}) - f(x_{\pi(m)}) = \sum_{i=1}^{m-1} f(x_{\pi(i)}) - f(x_{\pi(i+1)})$$

where the first inequality uses the fact that $f(x_{\pi(i)})$ is non-increasing in $\pi(i)$ and $f(x_{\pi(i)}) \geq f_\star$ and the equality is simply the definition of the telescoping sum of $f(x_{\pi(m)}) - f(x_{\pi(1)})$. Therefore,

$$\Delta_f \geq \sum_{i=1}^{m-1} f(x_{\pi(i)}) - f(x_{\pi(i+1)}) = \sum_{i=1}^{m-1} \hat{\rho}_{\pi(i)} \left( -M_k(d_{\pi(i)}) + \frac{\theta}{2} \|\nabla f(x_{\pi(i)} + d_{\pi(i)})\| \|d_{\pi(i)}\| \right)$$

$$\geq \sum_{i=1}^{m-1} \beta \left( -M_k(d_{\pi(i)}) + \frac{\theta}{2} \|\nabla f(x_{\pi(i)} + d_{\pi(i)})\| \|d_{\pi(i)}\| \right) \geq \frac{\beta\theta}{2} \sum_{i=1}^{m-1} \|\nabla f(x_{\pi(i)} + d_{\pi(i)})\| \|d_{\pi(i)}\|$$

$$\geq \frac{\epsilon\beta\theta}{2} (m - 1) \underline{d}_\epsilon$$

where the first equality uses the definition of $\hat{\rho}_{\pi(i)}$, the second inequality follows from $\hat{\rho}_{\pi(i)} \geq \beta$ for $\pi(i) \in \mathcal{P}_\epsilon$, the third inequality uses that $-M_k(d_{\pi(i)}) \geq 0$, the final inequality uses that $\pi(i) \in \mathcal{P}_\epsilon$ implies that $\|\nabla f(x_{\pi(i)} + d_{\pi(i)})\| \geq \epsilon$ (by definition of $\pi(i) \in \mathcal{P}_\epsilon$) and $\underline{d}_\epsilon \leq \|d_{\pi(i)}\|$ (due to Lemma 4).

Rearranging the latter inequality for $m$ using the fact that $\beta\theta\epsilon\underline{d}_\epsilon > 0$ and $\Delta_f \geq 0$ yields $m \leq \frac{2\Delta_f}{\beta\theta\epsilon\underline{d}_\epsilon} + 1 = \frac{\bar{d}_\epsilon}{\underline{d}_\epsilon\omega} + 1 =$ where the equalities use the definitions of $\bar{d}_\epsilon$ and $\underline{d}_\epsilon$. $\qquad\square$

### A.3 Proof of Theorem 1

*Proof.* Define:

$$n_j := |\{k \in \mathbf{N} : k \notin \mathcal{P}_\epsilon, k < K_\epsilon, \underline{k}_\epsilon < k \leq j\}|$$
$$p_j := |\{k \in \mathcal{P}_\epsilon : \underline{k}_\epsilon < k \leq j\}|.$$

First we establish that

$$n_\infty \leq p_\infty + \log_\omega\left(\max\left\{\frac{\bar{d}_\epsilon}{\underline{d}_\epsilon}, 1\right\}\right). \tag{14}$$

Consider the induction hypothesis that

$$r_k \leq r_{\underline{k}_\epsilon}\omega^{p_k - n_k} \quad \forall k \in [\underline{k}_\epsilon, K_\epsilon) \cap \mathbf{N}. \tag{15}$$

If $k = \underline{k}_\epsilon$ then $p_k = n_k = 0$ and the hypothesis holds. Suppose that the induction hypothesis holds for $k = j$. Note that for all $j \in \mathbf{N}$ either $p_{j+1} = p_j + 1$ (and $n_{j+1} = n_j$) or $n_{j+1} = n_j + 1$ (and $p_{j+1} = p_j$). If $p_{j+1} = p_j + 1$ then

$$r_{j+1} = \|d_j\|\omega \leq r_j\omega \leq r_{\underline{k}_\epsilon}\omega^{p_j - n_j + 1} = r_{\underline{k}_\epsilon}\omega^{p_{j+1} - n_{j+1}}.$$

On the other hand, if $n_{j+1} = n_j + 1$ then

$$r_{j+1} = \|d_j\|/\omega \leq r_j/\omega \leq r_{\underline{k}_\epsilon}\omega^{p_j - n_j - 1} = r_{\underline{k}_\epsilon}\omega^{p_{j+1} - n_{j+1}}.$$

Therefore by induction (15) holds. By (15) and Lemma 4,

$$\underline{d}_\epsilon \leq \bar{d}_\epsilon\omega^{p_k - n_k}$$

which establishes (14).

By Lemma 4 we have $\underline{k}_\epsilon \leq 1 + \log_{\gamma_2\omega}(\max\{1, \underline{d}_\epsilon/r_1, r_1/\bar{d}_\epsilon\})$ and Lemma 5 we have $p_\infty \leq \frac{\bar{d}_\epsilon}{\underline{d}_\epsilon\omega} + 1$; using these inequalities in conjuction with (14) gives

$$K_\epsilon = \underline{k}_\epsilon + p_\infty + n_\infty + 1 \leq \underline{k}_\epsilon + 2p_\infty + \log_\omega\left(\max\{\bar{d}_\epsilon/\underline{d}_\epsilon\}\right) + 1$$

$$\leq \log_{\omega\gamma_2}(\max\{1, \underline{d}_\epsilon/r_1, r_1/\bar{d}_\epsilon\}) + \frac{2\bar{d}_\epsilon}{\underline{d}_\epsilon\omega} + \log_\omega(\max\{1, \bar{d}_\epsilon/\underline{d}_\epsilon\}) + 3$$

$$\leq \frac{2\bar{d}_\epsilon}{\underline{d}_\epsilon\omega} + 2\log_{\omega\gamma_2}\left(\max\left\{\frac{\bar{d}_\epsilon}{\underline{d}_\epsilon}, \frac{\underline{d}_\epsilon}{r_1}, \frac{r_1}{\bar{d}_\epsilon}, 1\right\}\right) + 3$$

$$= c_2 \cdot \frac{\Delta_f L^{1/2}}{\epsilon^{-3/2}} + 2\log_{\omega\gamma_2}\left(\max\left\{\frac{c_2\omega}{2} \cdot \frac{\Delta_f L^{1/2}}{\epsilon^{3/2}}, \frac{\gamma_2}{\omega c_1^{1/2}} \cdot \frac{\epsilon^{1/2}}{L^{1/2}r_1}, \frac{\beta\theta}{2\omega} \cdot \frac{r_1 L^{1/2}}{\epsilon^{1/2}}, 1\right\}\right) + 3$$

where

$$c_2 := \frac{4c_1^{1/2}\omega}{\beta\theta\gamma_2}$$

is a problem-independent constant. As $c_1, c_2, \omega, \beta, \theta, \gamma_1, \gamma_2$ and $\gamma_3$ are problem-independent constants (see the definition of $c_1$ in Lemma 2 and the requirements of Algorithm 1) the result follows. $\qquad\square$

## B  Proof of Theorem 2

We first prove Theorem 3 and then reduce Theorem 2 to Theorem 3. The following fact will be useful.

**Fact 3** ([53]). *If $f$ is $\alpha$-strongly convex and $S$-smooth on the set $C$ (i.e., $\alpha\mathbf{I} \preceq \boldsymbol{\nabla}^2 f(x) \preceq S\mathbf{I}$ for all $x \in C$) then*

$$\alpha\|x - x_\star\| \le \|\boldsymbol{\nabla} f(x)\| \le S\|x - x_\star\| \tag{16}$$

*where $x_\star$ is any minimizer of $f$.*

**Theorem 3.** *Suppose that $f$ is $L$-Lipschitz, $\boldsymbol{\nabla} f(x_\star) = 0$ and there exists $\alpha, S, t > 0$ such that $\alpha\mathbf{I} \preceq \boldsymbol{\nabla}^2 f(x) \preceq S\mathbf{I}$ for all $x \in \{x \in \mathbf{R}^n : \|x - x_\star\| \le t\}$. Consider the set*

$$C := \left\{ x \in \mathbf{R}^n : f(x) \le f(x_\star) + \frac{2\eta^2}{\alpha}, \|x - x_\star\| \le \eta \right\}$$

*with*

$$\eta = \min\left\{ t, \frac{\alpha^3(1-\gamma_1)}{2LS^2} \min\left\{\frac{1}{2}, \omega\gamma_2 - 1\right\}, \frac{12(1-\beta)\alpha}{L\omega\gamma_2}, \frac{\beta\theta(1-\beta)\alpha}{4\omega\gamma_2 Lc_1} \right\}$$

*then if $x_i \in C$ then for $k \ge 2 + i + \log_{\gamma_2\omega}(\frac{\eta}{\|d_i\|})$ we have*

$$\|x_{k+1} - x_\star\| \le \frac{2LS^2}{\alpha^3(1-\gamma_1)}\|x_k - x_\star\|^2.$$

*Proof.* We begin by establishing the premise of Lemma 6. First we establish $x_k \in C \implies x_{k+1} \in C$. Suppose that $x_k \in C$ then $f(x_{k+1}) \le f(x_k) \le f(x_\star) + \frac{2\eta^2}{\alpha}$. By strong convexity we get $x_{k+1} \in C$. Next we establish that $\min\{\gamma_2 r_k, \|x_{k+1} - x_\star\|\} \le \|d_k\| \le \omega\gamma_2\|x_k - x_\star\|$. By strong convexity and (6d) we have

$$\frac{\alpha + \delta_k}{2}\|d_k\|^2 - \|\boldsymbol{\nabla} f(x_k)\|\|d_k^N\| \le M_k(d_k^N) \le 0$$

which implies $\|d_k\| \le \frac{2\|\boldsymbol{\nabla} f(x_k)\|}{\alpha+\delta_k}$. Furthermore, by (9), (6a) and $\|d_k\| \le \frac{2\|\boldsymbol{\nabla} f(x_k)\|}{\alpha+\delta_k}$ we have

$$\|\boldsymbol{\nabla} f(x_k + d_k) + \delta_k d_k\| \le \|\boldsymbol{\nabla} M_k(d_k) + \delta_k d_k\| + \frac{L}{2}\|d_k\|^2 \le \gamma_1\|\boldsymbol{\nabla} f(x_k + d_k)\| + \frac{2L\|\boldsymbol{\nabla} f(x_k)\|^2}{\alpha^2}$$

which after rearranging

$$\|\boldsymbol{\nabla} f(x_k + d_k) + \delta_k d_k\| \le \frac{2L}{\alpha^2(1-\gamma_1)}\|\boldsymbol{\nabla} f(x_k)\|^2 \tag{17}$$

By strong convexity and smoothness,

$$\|x_k + d_k - \hat{x}_k\| \le \frac{2LS^2}{\alpha^3(1-\gamma_1)}\|x_k - x_\star\|^2 \tag{18}$$

where $\hat{x}_k := \min f(x) + \frac{\delta_k}{2}\|x - x_k\|^2$. Therefore, as $\|x_k - x_\star\| \le \frac{\alpha^3(1-\gamma_1)}{2LS^2} \min\left\{\frac{1}{2}, \omega\gamma_2 - 1\right\}$,

$$\|x_k + d_k - \hat{x}_k\| \le \min\left\{\frac{1}{2}, \omega\gamma_2 - 1\right\}\|x_k - x_\star\|$$

which combined with the triangle inequality and $\|\hat{x}_k - x_k\| \le \|x_k - x_\star\|$ gives

$$\|d_k\| \le \|x_k + d_k - \hat{x}_k\| + \|x_k - \hat{x}_k\| \le \omega\gamma_2\|x_k - x_\star\|$$

Furthermore, if $\|d_k\| < \gamma_2 r_k$ then by (6b) we have $\delta_k = 0$ and $\hat{x}_k = x_\star$ which gives

$$\|x_k + d_k - x_\star\| \le \frac{1}{2}\|x_k - x_\star\| \le \|x_k - x_\star\| - \|x_k + d_k - x_\star\| \le \|d_k\|.$$

Next we show $x_k \in C$ implies $\hat{\rho}_k \ge \beta$. To obtain a contradiction we assume $\hat{\rho}_k < \beta$, by the definition of the model, (6a) and strong convexity we get

$$M_k(d_k) = \frac{1}{2}d_k^T\boldsymbol{\nabla}^2 f(x_k)d_k + \boldsymbol{\nabla} f(x_k)^T d_k = d_k^T(\boldsymbol{\nabla}^2 f(x_k)d_k + \delta_k d_k + \boldsymbol{\nabla} f(x_k)) - \frac{1}{2}d_k^T(\boldsymbol{\nabla}^2 f(x_k) + 2\delta_k\mathbf{I})d_k$$

$$\le \gamma_1\|d_k\|\|\boldsymbol{\nabla} f(x_{k+1})\| - \frac{1}{2}d_k^T(\boldsymbol{\nabla}^2 f(x_k) + 2\delta_k\mathbf{I})d_k$$

$$\le \gamma_1\|d_k\|\|\boldsymbol{\nabla} f(x_{k+1})\| - \frac{\alpha}{2}\|d_k\|^2.$$

It follows that by inequality (10), $\|d_k\| \le \omega\gamma_2\|x_k - x_\star\| \le \frac{12}{L}(1-\beta)\alpha$, inequality (11), $\|d_k\| \le \omega\gamma_2\|x_k - x_\star\| \le \frac{\beta\theta(1-\beta)\alpha}{4Lc_1}$ we have

$$
\begin{aligned}
f(x_k) - f(x_{k+1}) &\ge -\beta M_k(d_k) + \frac{(1-\beta)\alpha}{2}\|d_k\|^2 - \frac{L}{6}\|d_k\|^3 \\
&\ge -\beta M_k(d_k) + \frac{(1-\beta)\alpha}{4}\|d_k\|^2 \\
&\ge -\beta M_k(d) + \frac{(1-\beta)\alpha}{4Lc_1}\|\boldsymbol{\nabla} f(x_k)\| \\
&\ge -\beta M_k(d) + \beta\theta\|\boldsymbol{\nabla} f(x_k)\|\|d_k\|
\end{aligned}
$$

which gives our desired contradiction.

With the premise of Lemma 6 established we conclude that for $k \ge 2 + i + \log(\eta/\|d_i\|)$ we have $\delta_k = 0$ and therefore by (18) we get the desired result. $\qquad\square$

The following Lemma is a standard result but we include it for completeness.

**Lemma 7.** *If $\boldsymbol{\nabla}^2 f(x_\star)$ is twice differentiable and positive definite, then there exists a neighborhood $N$ and positive constants $\alpha, \beta > 0$ such that $\alpha\mathbf{I} \preceq \boldsymbol{\nabla}^2 f(x) \preceq S\mathbf{I}$ for all $x \in N$.*

*Proof.* As $\boldsymbol{\nabla}^2 f$ is twice differentiable and the fact that continuous functions on compact sets are bounded we conclude that there exists a neighborhood $N$ around $x_\star$ that $\boldsymbol{\nabla}^2 f$ is $L$-Lipschitz for some constant $L \in (0, \infty)$. Then by using the fact that there exists positive constants $\alpha', \beta' \in (0, \infty)$ s.t. $\alpha'\mathbf{I} \preceq \boldsymbol{\nabla}^2 f(x_\star) \preceq \beta'\mathbf{I}$ we conclude for sufficiently small ball around $x_\star$ we have $\alpha'/2\mathbf{I} \preceq \boldsymbol{\nabla}^2 f(x) \preceq 2\beta'\mathbf{I}$ for all $x$ in a sufficiently small neighborhood $N' \subseteq N$. $\qquad\square$

*Proof of Theorem 2.* Follows by Lemma 7 and Theorem 3. $\qquad\square$

## C  Solving trust-region subproblem

In this section, we detail our approach to solve the trust-region subproblem. We first attempt to take a Newton's step by checking if $\boldsymbol{\nabla}^2 f(x_k) \succeq 0$ and $\|\boldsymbol{\nabla}^2 f(x_k)^{-1}\boldsymbol{\nabla} f(x_k)\| \le r_k$. However, if that is not the case, then the optimally conditions mentioned in (6), will be a key ingredient in our approach to find $\delta$ and hence $d_k(\delta)$. Based on these optimally conditions, we will define a univariate function $\phi$ that we seek to find its root at each iteration. In our implementation we use $\gamma_3 = 1.0$ for (6d) which is the same as satisfying (5d). The function $\phi$ is defined as bellow:

$$
\phi(\delta) := \begin{cases} -1, & \text{if } \boldsymbol{\nabla}^2 f(x_k) + \delta\mathbf{I} \not\succeq 0 \text{ or } \|d_k(\delta)\| > r_k \\ +1, & \text{if } \boldsymbol{\nabla}^2 f(x_k) + \delta\mathbf{I} \succeq 0 \ \& \ \|d_k(\delta)\| < \gamma_2 r_k \\ 0, & \text{if } \boldsymbol{\nabla}^2 f(x_k) + \delta\mathbf{I} \succeq 0 \ \& \ \|d_k(\delta)\| \le r_k \end{cases}
$$

where:

$$
d_k(\delta) := (\boldsymbol{\nabla}^2 f(x_k) + \delta\mathbf{I})^{-1}(-\boldsymbol{\nabla} f(x_k))
$$

When we fail to take a Newton's step, we first find an interval $[\delta, \delta']$ such that $\phi(\delta) \times \phi(\delta') \le 0$. Then we apply bisection method to find $\delta_k$ such that $\phi(\delta_k) = 0$. In case our root finding logic failed, then we use the approach from the hard case section under chapter 4 "Trust-Region Methods" in [44] to find the direction $d_k$.

The logic to find the interval $[\delta, \delta']$ is summarized as follow. We first compute $\phi(\delta)$ using the $\delta$ value from the previous iteration. Then we search for $\delta'$ by starting with $\delta' = 2\delta$. We compute $\phi(\delta')$ and in the case $\phi(\delta') < 0$, we update $\delta'$ to become twice its current value, otherwise if $\phi(\delta') > 0$, we update $\delta'$ to become half its current value. We keep repeating this logic until we get a $\delta'$ such that $\phi(\delta) \times \phi(\delta') \le 0$ or until we reach the maximum iteration limit which is marked as a failure.

The whole approach is summarized in Algorithm 2:

**Algorithm 2:** trust-region subproblems solver

---

**if** $\nabla^2 f(x_k) \succeq 0$ **then**
  $\quad d_k = -\nabla^2 f(x_k)^{-1} \nabla f(x_k)$
  $\quad$**if** $\|d_k\| \leq r$ **then**
  $\quad\quad$**return** $d_k$;
**if** *hard case* **then**
  $\quad$Find $d_k$ using [44, pages 87-88] ;
  $\quad$**return** $d_k$
**else**
  $\quad$Find initial interval $[\delta, \delta']$ using the $\phi$ function such that $\phi(\delta) \times \phi(\delta') \leq 0$ ;
  $\quad$Use bisection method to find $\delta_k$ such that $\phi(\delta_k) = 0$ ;
  $\quad$**return** $d_k(\delta_k)$

---

## D    Experimental results details

### D.1    Learning linear dynamical systems

The time-invariant linear dynamical system is defined by:

$$h_{t+1} = Ah_t + Bu_t + \xi_t$$
$$x_t = h_t + \vartheta_t$$

where the vectors $h_t$ and $x_t$ represent the hidden and observed state of the system at time $t$. Here $u_t, \vartheta_t \sim N(0,1)^d$, $\xi_t \sim N(0,\sigma)^d$ and $A$ and $B$ are linear transformations.

The goal is to recover the parameters of the system using maximum likelihood estimation and hence we formulate the problem as follow:

$$\min_{A,B,h} \sum_{t=1}^{T} \frac{\|h_{t+1} - Ah_t - Bu_t\|^2}{\sigma^2} + \|x_t - h_t\|^2$$

We synthetically generate examples with noise both in the observations and also the evolution of the system. The entries of the matrix $B$ are generated using a Normal distribution $N(0,1)$. For the matrix $A$, we first generate a diagonal matrix $D$ with entries drawn from a uniform distribution $U[0.9, 0.99]$ and then we construct a random orthogonal matrix $Q$ by randomly sampling a matrix $W \sim N(0,1)^{d \times d}$ and then performing an QR factorization. Finally using the matrices $Q$ and $D$, we define $A$:

$$A = Q^T DQ$$

We compare our method against the Newton trust-region method available through the Optim.jl package [51] licensed under `https://github.com/JuliaNLSolvers/Optim.jl/blob/master/LICENSE.md`. In the results/learning problem subdirectory in the git repository, we present the full results of running our experiments on 60 randomly generated instances with $T = 50$, $d = 4$, and $\sigma = 0.01$ where we used a value of $10^{-5}$ for the gradient termination tolerance. This experiment was performed on a MacBook Air (M1, 2020) with 8GB RAM.

### D.2    Matrix completion

The original power consumption data is denoted by a matrix $D \in R^{n_1 \times n_2}$ where $n_1$ represents the number of measurements taken per day within a 15 mins interval and $n_2$ represents the number of days. Part of the data is missing, hence the goal is to recover the original data. The set $\Omega = \{(i,j)|D_{i,j} \text{ is observed}\}$ denotes the indices of the observed data in the matrix $D$.

We decompose $D$ as a product of two matrices $P \in R^{n_1 \times r}$ and $Q \in R^{n_2 \times r}$ where $r < n_1$ and $r < n_2$:

$$D = PQ^T.$$

To account for the effect of time and day on the power consumption data , we use a baseline estimate [54]:

$$d_{i,j} = \mu + r_i + c_j$$

where $\mu$ denotes the mean for all observed measurements, $r_i$ denotes the observed deviation during time $i$, and $c_j$ denotes the observed deviation during day $j$ [49, 54].

We formulate the matrix completion problem as the regularized squared error function of SVD model [49, Equation 10]:

$$\min_{r,c,p,q} \sum_{(i,j)\in\Omega} (D_{i,j} - \mu - r_i - c_j - p_i q_j^T) + \lambda_1(r_i^2 + c_j^2) + \lambda_2(\|p_i\|_2^2 + \|q_j\|_2^2)$$

We use the public data set of Ausgrid, but we only use the data from a single substation (the Newton trust-region method [51] is very slow for this example so testing it on all substations takes a prohibitively long time). We limit our option to 30 days and 12 hours measurements i.e the matrix D is of size $48 \times 30$ because with a larger matrix size, the Newton trust-region [51] was always reaching the iterations limit.

We compare our method against Newton trust-region algorithm available through the Optim.jl package [51] licensed under `https://github.com/JuliaNLSolvers/Optim.jl/blob/master/LICENSE.md`. In the results/matrix completion subdirectory in the git repository, we include the full results of running our experiments on 10 instances by randomly generating the sampled measurements from the matrix $D$ with the same values for the regularization parameters as in [49] where we used a value of $10^{-5}$ for the gradient termination tolerance. This experiment was performed on a MacBook Air (M1, 2020) with 8GB RAM.