# OpenReview forum: "A consistently adaptive trust-region method"
_NeurIPS.cc/2022/Conference — NeurIPS 2022 Accept_

### Official Review · Reviewer_VLHh · 2022-07-05

**Rating:** 6
**Confidence:** 3
**Soundness:** 3 good
**Presentation:** 3 good
**Contribution:** 3 good

**Summary:**

The paper presents an adaptive trust method region whose complexity in terms of iterations matches that of non adaptive trust region methods. Crucially, this method has the right exponents in terms of the Lipschitz constant of the Hessian and the error.

More specifically, the introduction of a factor controlling the size of the direction and gradient in the update rule allows to not pay the regularisation parameter and to control more finely the different stages of the algorithm.

The proof of convergence towards and epsilon approximate stationary point is divided into different phases.

Additionally, a proof of quadratic convergence near a local optimum is given.

Experiments to support the fact that this method compares with existing methods while having better theoretical guarantees are given,

**Questions:**


- I believe in lemma 2, typo between $\gamma_1$ and $\gamma_2$ in the second term of the constant (first bound in the appendix). Also, another typo is that lemma 2 applies if the inequality is strict, while in the next proofs it is used for large inequality (but OK since in the algorithm the case distinction implies that the inequality is strict.
- In the proof of lemma 5, I believe the third term should be an inequality and not an equality ?
- Are there cases where ARC fails because of the sigma problem, where you could show that your method is better ?


**Limitations:**

Exposed by the authors.

**Strengths And Weaknesses:**

    * Originality:

The method is new in the introduction of the term controlling the size of $d_k$ in this context.



    * Quality: I read most of the proofs and they are sound. Globally, I think the paper is very thorough. One thing missing from the main paper is perhaps how to get the trust region solution to satisfy 7 (it is mentioned in the references and in the appendices) but I am not a specialist.



    * Clarity:

Relatively clear. I just believe the part about $\rho_k < \beta$ or $>= \beta$ should be made slightly clearer in the discussion about the different phases. Otherwise, super clear.




    * Significance: For me, the work is significant theoretically, and is backed by good experiments.

---

> ### Author Response · Authors · 2022-08-02
> **Response to reviewer comments**
>
> *I just believe the part about $\rho_k < \beta$ or $\ge \beta$ should be made slightly clearer in the discussion about the different phases.*
>
> We presume you are referring to line 105. We will rewrite that sentence to make it clearer.
>
> *I believe in lemma 2, typo between $\gamma_1$ and $\gamma_2$ in the second term of the constant (first bound in the appendix).*
>
> Agreed. Thank you for the correction.
>
> *Also, another typo is that lemma 2 applies if the inequality is strict, while in the next proofs it is used for large inequality (but OK since in the algorithm the case distinction implies that the inequality is strict.*
>
> Could you clarify what you mean? We presume you are referring to the inequality $|| d_k || < \gamma_2 r_k$ but we are not sure where you think the proof of Lemma 3 goes wrong.
>
> *In the proof of lemma 5, I believe the third term should be an inequality and not an equality ?*
>
> This term should be an equality. We are just rewriting the same sum in a different way. However, we agree that the current proof is somewhat confusing. Indeed this final transition is not really necessary if one uses the second term in the remainder of the proof. We will improve the proof accordingly.
>
> *Are there cases where ARC fails because of the sigma problem, where you could show that your method is better ?*
>
> We believe that the hard example for trust-region methods given by [40] (also, see Figure 3b) can be modified to “stall” ARC but the modification required would be very convoluted. Indeed the example given in [40] is already quite involved.

---

### Official Review · Reviewer_mz9Z · 2022-07-15

**Rating:** 7
**Confidence:** 3
**Soundness:** 3 good
**Presentation:** 4 excellent
**Contribution:** 3 good

**Summary:**

This manuscript studies adaptive trust-region methods and makes an argument that a large bulk of existing literature has an underlying dependency on an arbitrarily small parameter that renders available bounds vacuous. Moreover, dependency on problem parameters is suboptimal, e.g., the Lipschitz constants. The result lies in a modified threshold or ratio for the search direction. Quadratic convergence is shown when the iterates are sufficiently close to the optimal point. Some numerical results are provided.

**Questions:**

Can the developed techniques be applied to other adaptive methods beyond cubic regularization?

Is there a function class for which the advantage of the proposed method is more evident than existing approaches?

**Limitations:**

No potential negative societal impact

**Strengths And Weaknesses:**

Strengths
- A weakness of existing methods is clearly identified and solved.
- The originality lies in the definition of new search parameters.
- The paper is well written and the result can serve as the base of future work in adaptive methods.
- Some minimal numerical results are shown

Weaknesses
- At the core of the main argument is the suboptimality of existing approaches. However, the discussion and subsequent argument about it are relegated to a footnote, with very limited discussion.
- Contributions 2 and 4 do not read as contributions.
- The numerical section is very limited. The argument is that the provided method is better than the literature but the provided numerical evidence only explores some minimal examples rather than a comprehensive exposition of the computational advantages.

---

> ### Author Response · Authors · 2022-08-02
> **Response to other reviewer comments**
>
> *At the core of the main argument is the suboptimality of existing approaches. However, the discussion and subsequent argument about it are relegated to a footnote, with very limited discussion.*
>
> Yes, we will expand the discussion of the suboptimality of existing approaches significantly in the revision.
>
> *Contributions 2 and 4 do not read as contributions.*
>
> Thank you for the feedback, we will rewrite accordingly.
>
> *The argument is that the provided method is better than the literature but the provided numerical evidence only explores some minimal examples rather than a comprehensive exposition of the computational advantages.*
>
> We only claim that (line 14-16) “Our method ... in our experiments performs **competitively** with both ARC and a classical trust-region method”. We do not claim that the method is practically better. Indeed, compared with a classic trust-region method we only expect differences on convoluted worst-case instances (e.g., Figure 3(b)). Our main contribution is that we have strong worst-case guarantees without sacrificing practical performance.
>
> *Can the developed techniques be applied to other adaptive methods beyond cubic regularization?*
>
> I think you mean trust-region rather than cubic regularization. We believe so, but do not feel comfortable discussing our ideas for potential follow up papers in a public forum.
>
> *Is there a function class for which the advantage of the proposed method is more evident than existing approaches?*
>
> In practice, this method is very similar to a classic trust-region method, our change from $\rho_k$ to $\hat{\rho}_k$ has little impact on practical performance (see Figure 3(a) and discussion on lines 256-267). Theoretically, it does give a much better complexity than a classic trust-region method (indeed classic trust-region has an $\Omega(\epsilon^{-2})$ complexity as we describe on Line 262).

---

> > ### Comment · Reviewer_mz9Z · 2022-08-10
> > **Satisfactory answers**
> >
> > The reviewer thanks the authors for the responses, I think the contribution is clear, I changed my score accordingly.

---

> ### Author Response · Authors · 2022-08-02
> **Response to "The numerical section is very limited"**
>
> In defense of the number of examples of our numerical results, in addition to the learning linear dynamical system problem (which we average across 60 random seeds), we currently run 67 instances from the CUTEst test set and aggregate the results (Figure 2).
>
> Nonetheless, to supplement our numerics we will add some matrix completion results. We will use the formulation of Zuhang et al. [2020] from the paper “Power Load Data Completion Method Considering Low Rank Property''. The matrix completion formulation can be written as the regularized squared error function of SVD model which is stated in equation 10 in their paper. Similar to their experiment, we use the public data set of Ausgrid, but we only use the data from a single substation (the Newton trust region method [45] is very slow for this example so testing it on all substations takes a prohibitively long time). The load data is represented by a matrix D of size 30 * 48 where the number of rows represent the number of days and the number of columns represent the number of measurements taken per day within a 15 mins interval.  We limit our option to these values because with a larger matrix size, the Newton trust region [45] was always reaching the iterations limit. We use the same values for the regularization parameters as in their paper. Also we set the iteration limit to $1000$ and the gradient termination tolerance to $10^{-5}$. We perform 10 experiments by randomly generating the sampled measurements from the matrix D and average the results:
>
> |      | #iter |
> | ----------- | ----------- |
> | Newton trust-region [45]    | 1000      |
> | Our method    | 218.812      |
>
> Note the Newton trust-region [45] is hitting the iteration limit for all $10$ problems.

---

### Official Review · Reviewer_ctCf · 2022-07-28

**Rating:** 6
**Confidence:** 3
**Soundness:** 3 good
**Presentation:** 4 excellent
**Contribution:** 3 good

**Summary:**

This paper proposes a new adaptive variant of the classic trust region method, named FLAT. The proposed scheme preserves the strong convergence guarantee without requiring the knowledge of problem dependent constants like the Lipschitz constant. To achieve a first order stationary point, when compared with previous adaptive second order methods, FLAT enjoys a better depends on the Lipschitz constant and removes the dependence on a regularization parameter which potentially makes the previous guarantees vacuous.

**Questions:**

It would be great if the authors can comment on the intuition of the newly constructed adaptive rule in Eq. (9). It seems that the additional term $||\nabla f(x_k + d_k)||\cdot||d_k||$ is specifically tailored to quickly achieve a first order stationary point. Is this the reason why the second order guarantee of FLAT is absent? Is it possible to revise the adaptive rule to also include the second order information in order to establish the second order guarantee?

**Limitations:**

There is no potential negative societal impact.

As a second order method, the proposed method lacks the guarantee of converging to the second order stationary point. This drawback is acknowledged by the authors as well.

**Strengths And Weaknesses:**

Strengths:

This paper is well written and the major contribution of this work is well articulated: The proposed FLAT method, as an adaptive method, is able to achieve the same dependence on key parameters $(L, \epsilon)$ as methods that have the exact information of the Lipschitz constant of the Hessian. Moreover, it removes the dependence on $1/\sigma_\min$ which is a potentially unbounded quantity present in previous approaches.

Weaknesses:

The weakness of this work is also pretty clear: As a second order method, it lacks the guarantee of converging to the second order stationary point which is the key advantage when compared against the significantly more efficient first order method. This drawback is acknowledged by the authors as well.

Also, the experiments in Figure 2. seems to suggest that ARC has a better performance than FLAT, in all regards.

---

> ### Author Response · Authors · 2022-08-02
> **Author response to reviewer questions**
>
> *It would be great if the authors can comment on the intuition of the newly constructed adaptive rule in Eq. (9). It seems that the additional term $|| \nabla f(x_k+d_k) || \cdot \|| d_k \||$ is specifically tailored to quickly achieve a first order stationary point.*
>
> Some limited intuition is given at scatter points in the paper. For example, see Lines 111-113, 122-128, 181-184. The cleanest intuition we (as given on Line 181-814) is that one can show $|| d_k ||$ is $\Omega( L^{½} \epsilon^{½} )$ which means the function is reduced by $\Omega( L^{½} \epsilon^{3/2} )$ at each successful iteration.
>
> *Is this the reason why the second order guarantee of FLAT is absent?*
>
> Yes, this is the reason the second-order guarantee is absent. Indeed, we believe that one cannot prove a second-order guarantee for FLAT.
>
> *Is it possible to revise the adaptive rule to also include the second order information in order to establish the second order guarantee?*
>
> Yes, this is likely possible. This is an interesting topic for future research.

---

### Author Response · Authors · 2022-08-02
**Reframing contributions after experts pointed out missing papers**

First, we want to thank the reviewers for their feedback.

After submitting this paper to Neurips we shared it with some experts on this topic. They gave us the following useful feedback that we wish to share with the reviewers. Importantly, this feedback means we need to reframe our contributions. While we feel our reframed contributions are still significant, we understand reviewers may wish to reevaluate in light of this new information.

The first piece of (minor) feedback is that a better term than "fully adaptive" is "consistently adaptive", as this phrase more accurately describes the theoretical property it defines. We will consequently rename the algorithm CAT (**C**onsistently **A**daptive **T**rust-Region Method).

Most critically, they pointed us to two important papers we had missed:
1) Cartis, Coralia, Nick I. Gould, and Philippe L. Toint. Universal regularization methods: varying the power, the smoothness and the accuracy. SIAM Journal on Optimization, (2019).
2) Regularized Newton Methods for Minimizing Functions with Hölder Continuous Hessians. G. N. Grapiglia and Yu. Nesterov. SIAM Journal on Optimization, (2017).

These two papers require us to reframe our contributions. In particular, these methods are consistently adaptive. However, they are based on cubic regularized Newton methods rather than trust-region methods. Therefore, it remains the case that prior to our work there was no existing consistently adaptive trust-region method. Also, it is unclear if these papers represent practical methods as neither paper provides numerical results. Nonetheless we will update the paper accordingly. The following paragraph shows how we intend to update the abstract.

*Adaptive trust-region methods attempt to maintain strong convergence guarantees without depending on conservative estimates of problem properties such as Lipschitz constants. However, on close inspection, one can show existing adaptive trust-region methods have theoretical guarantees with severely suboptimal dependence on problem properties such as the Lipschitz constant of the Hessian. For example, TRACE developed by Curtis et al. obtains a $O(\Delta_f L^{3/2} \epsilon^{-3/2}) + \tilde{O}(1)$ iteration bound where $L$ is the Lipschitz constant of the Hessian. Compared with the optimal $O(\Delta_f L^{1/2} \epsilon^{-3/2})$ bound this is suboptimal with respect to $L$. We present the first adaptive trust-region method which circumvents this issue and requires at most $O( \Delta_f L^{1/2}  \epsilon^{-3/2}) + \tilde{O}(1)$ iterations to find an $\epsilon$-approximate stationary point, matching the optimal iteration bound up to an additive logarithmic term. Our method is a simple variant of a classic trust-region method and in our experiments performs competitively with both ARC and a classical trust-region method.*

---

> ### Author Response · Authors · 2022-08-02
> **Adding to trust-region literature after expert feedback**
>
> Another piece of feedback they gave is that we did not discuss the trust-region literature in sufficient detail, particularly in the introduction. Therefore, we will add a paragraph along these lines to the introduction:
>
> *There is an extensive literature on trust-region methods [9,10,22,25,27,A,B,C,D] with several works considering their worst-case iterations bounds [35,36,40] but there are no consistently adaptive methods. Classic trust-region methods are not consistently adaptive, indeed, in the worst-case they require a number of iterations proportional to $\epsilon^{-2}$ [40]. [35,36] provides methods that adaptively controls the trust-region radius and achieve iteration bounds proportional to $\epsilon^{-3/2}$. However, neither method is consistently adaptive. In particular, inspection of both papers shows suboptimal scaling with respect to $L$ of $L^{2}$ for [35] and $L^{3/2}$ for [36] instead of the optimal scaling of $L^{½}$, where $L$ is the Lipschitz constant of the Hessian. Moreover, [36] is significantly more complicated than a classic trust-region method, even maintaining a cubic regularization parameter. On the other hand [35] offers a simpler method by adding a fixed quadratic regularization to each trust-region subproblem. However, this quadratic regularizer inhibits superlinear convergence$^\star$ and in practice slows down the method [35, Figure 1].*
>
> *$^\star$This can be seen by applying quadratically regularized Newton’s method to minimize $x^2$ with quadratic regularizer $\delta x^2$ one gets $x_k = x_0 (\delta / (\delta + 1))^k$ which is linear convergence for all $\delta > 0$.*
>
> Finally, they pointed out (as did Reviewer 2) that we should have presented the Footnote 2 in the main body and abstract, as ARC is usually run with the user choosing a $\sigma_{\min}$ value. The current way we describe things can be confusing to the reviewer.
>
> **Additional references:**
>
> A) Conn, Andrew R., Nicholas IM Gould, and Philippe L. Toint. Trust region methods. Society for Industrial and Applied Mathematics, 2000.
>
> B) ​​Powell, M. J. D. "On the global convergence of trust region algorithms for unconstrained minimization." Mathematical Programming 29.3 (1984): 297-303.
>
> C) Carter, Richard G. "On the global convergence of trust region algorithms using inexact gradient information." SIAM Journal on Numerical Analysis 28.1 (1991): 251-265.
>
> D) Powell, Michael JD. "On trust region methods for unconstrained minimization without derivatives." Mathematical programming 97.3 (2003): 605-623.

---

### Meta-Review · Area_Chair_SMex · 2022-08-22

**Recommendation:** Accept
**Confidence:** Less certain

**Metareview:**

The paper proposes a new adaptive trust region method. Because most of the reviewers think the paper is interesting, I recommend an acceptance.

**Award:**

No

---

### Decision · Program_Chairs · 2022-09-14

Accept